# One Health approach for *Brucella canis*: Serological and molecular detection in animal-hoarding individuals and their dogs

**Letícia Schiavo[1,2], Matheus Lopes Ribeiro[3], Meila Bastos de Almeida[2], Graziela Ribeiro da Cunha[4], Giselle Almeida Nocera Espírito Santo[2], Vivien Midori Morikawa[5], Acácia Ferreira Vicente[6], Claire Ponsart[6], Carlos Eduardo de Santi[7], Louise Bach Kmetiuk[8], Jane Megid** [3], **Alexander Welker Biondo** [1,7] *

**1** Graduate College of Cell and Molecular Biology, Federal University of Paraná (UFPR), Curitiba, Paraná, Brazil, **2** Paraná State Technology Institute (Tecpar), Paraná State Government, Curitiba, Paraná, Brazil, **3** Department of Animal Production and Preventive Veterinary Medicine, São Paulo State University (UNESP), Botucatu, São Paulo, Brazil, **4** Secretary of the Environment, City Hall of Pinhais, Pinhais, Paraná, Brazil, **5** Secretary of the Environment, City Hall of Curitiba, Curitiba, Paraná, Brazil, **6** WOAH/EU & National Reference Laboratory for Animal Brucellosis, Animal Health Laboratory, Paris-Est University/Anses, Paris, France, **7** Department of Veterinary Medicine, Federal University of Paraná (UFPR), Curitiba, Paraná, Brazil, **8** Zoonosis Surveillance Unit, City Secretary of Health, Curitiba, Paraná, Brazil

* abiondo@ufpr.br

**Data Availability Statement:** All relevant data are within the manuscript and its Supporting information files.

## Abstract

Animal hoarding disorder (AHD) is classified as a psychiatric obsessive-compulsive condition characterized by animal accumulation and often accompanied by unsanitary conditions and animal cruelty. Although AHD may increase pathogen transmission and spread, particularly for zoonotic diseases, human and dog exposure in such cases has yet to be fully established. Accordingly, this study aimed to assess *Brucella canis* in 19 individuals with AHD (11 households) and their 264 dogs (21 households) in Curitiba, the eighth largest city in Brazil, with approximately 1.8 million habitants. Anti-*B. canis* antibodies were detected by the 2-mercaptoethanol microplate agglutination test (2ME-MAT) and by a commercial lateral flow immunoassay (LFIA), while molecular detection of previously positive seropositive samples was performed by conventional PCR. Although all the human samples were 2ME-MAT negative, 12/264 (4.5%, 95% Confidence Interval: 2.0–7.0%) dog samples were 2ME-MAT and LFIA positive, with 2ME-MAT titers ranging from 20 to 640. At least one dog in 4/21 (19.0%, 95% CI: 2.0–46.0%) households was seropositive. Despite the absence of seropositivity in individuals with AHD and the comparatively low seroprevalence in dogs, *B. canis* circulation and outbreaks should be considered in such human populations due to the high burden and recurrent character of *B. canis* exposure in high-density dog populations and the constant introduction of susceptible animals.

## Author summary

Individuals with animal hoarding disorder (AHD), a psychiatric obsessive-compulsive disorder, are considered among the most vulnerable people due to their precarious

**Funding:** This study was supported by the Araucaria Foundation of Paraná state (grant number SUS2020111000010 to AWB). The funders had no role in study design, data collection and analysis, decision to publish, or preparation of the manuscript.

**Competing interests:** The authors have declared that no competing interests exist.

sanitary conditions and exposure to several zoonoses, including canine brucellosis, a neglected and underreported disease of public health concern worldwide. Close and continuous human contact with unhealthy and unassisted dogs may increase pathogen transmission and spread, which is worsened by elderly people's difficulty accessing health services and lack of hoarding control and management programs. The present study assessed *B. canis* as a silent pathogen in this hard-to-access vulnerable population, along with their dogs. Although all the individuals with AHD included herein were seronegative and their dogs presented relatively low seroprevalence, *B. canis* infection and outbreaks should always be considered, particularly due to unsanitary household conditions, the high density of dog populations, and the constant introduction of susceptible dogs of unknown origin. Thus, dogs seropositive for *B. canis* living in households of people with AHD should be considered a warning to local public health authorities. In conclusion, serological and molecular assessments of *B. canis* in people with AHD and other vulnerable populations may serve as instruments for effective public health policies, including diagnosis, control, monitoring, and prevention.

## Introduction

Animal hoarding disorder (AHD) is defined as a psychiatric obsessive-compulsive condition characterized by the accumulation of animals in households, frequently in high numbers and confined to small spaces and lacking minimum nutrition, physical space, sanitation, and veterinary assistance [1,2]. Individuals with AHD often live in unsanitary conditions, leading to vector proliferation and pathogen spread, particularly of zoonotic diseases [1,3–6]. In addition, such individuals often fail to recognize animal suffering, are often hungry, trapped, crowded, or even die [1], as indicated by case reports of dogs scavenging the owner's remains [7].

In a One Health approach, AHD should be considered beyond simple ownership of multiple pets, as this practice impacts the health, welfare and safety of owners themselves, their animals, their families, the environmental health of their households and the surrounding community [3,5].

Canine brucellosis, a zoonotic disease caused by the intracellular bacterium *B. canis* that has spread worldwide and is considered endemic in Brazil, is the main cause of infection in domestic dogs, mainly leading to reproductive failure and infertility, with reports of human infection [8–16]. *B. canis* may insidiously infect dogs and cause intermittent bacteremia for months or even years, causing dogs to become a continuous source of infection [13,17,18]. In such a scenario, dogs may become a potential risk for humans and other animals, in addition to posing an occupational risk for veterinarians, laboratory-handling personnel and breeders [12,19,20].

As *B. canis* may persist in prostate and lymphoid tissues, neutering may only prevent semen contamination and not eliminate infection [13,16,17,18]. Human infection has rarely been reported, partially due to the absence of investigations, with many endemic countries reporting "no data" and a lack of active surveillance to pinpoint the actual prevalence [12,15,17,21,22]. Clinical signs are nonspecific, transient, and often similar to those of influenza virus infection [16,17,19,21,23]. Concomitant detection of human and canine *B. canis* infections has been rare, as reported in Colombia and South America [11] and in case reports in New York City, USA [23]; Argentina, South America [24]; and the Netherlands [25].

Although markedly neglected, canine brucellosis has not been included in the list of mandatory notification diseases by the World Organization for Animal Health (WOAH) [14,26]. In

addition, underdiagnosis may occur due to low suspicion, as dog reproductive failure and abortion may not be considered primary signs for brucellosis diagnosis by veterinarians (and owners' physicians) [9,13,15].

The prevalence of dog anti-*B. canis* antibodies in Brazil has varied from 5/175 (2.9%) in Paraná state [27] to 7/106 (6.6%) in Rio de Janeiro state [28] and 16/254 (6.3%) to 39/254 (16.5%) in Minas Gerais and Espírito Santo states (depending on the test used) [29] to 23/32 (71.8%) in São Paulo state [30]. Despite the close contact of humans with dogs under poor sanitary conditions, the lack of veterinary and nutritional care and the susceptibility to zoonotic diseases [5], no study has been conducted among individuals with AHD and their companion dogs for *B. canis* infection. Accordingly, this study aimed to assess the seroprevalence of anti-*B. canis* antibodies and molecular *B. canis* detection in individuals with AHD and their dogs in Curitiba, the eighth largest Brazilian city, which has 1.8 million habitants.

## Materials and methods

### Ethics statement

This study was approved by the Animal Use Ethics Committee (protocol number 077/2015) and by the National Human Ethics Research Committee (protocol number 3,166,749/2019) through the Federal University of Paraná, southern Brazil.

### Study area

This study was conducted in Curitiba (25˚25'47" S and 49˚16'19" W), the capital of Paraná state and the eighth largest city in Brazil, which has more than 1.8 million habitants [31] and a high Municipal Human Development Index (MHDI) of 0.823, ranked 10[th] out of 5,568 Brazilian cities [31,32]. Fully covered by urban areas, with a subtropical climate and the highest and coldest Brazilian capital (among 27 total), Curitiba has an average annual temperature of 16.5˚C and an altitude of 945 m above sea level [32].

### Sample collection

As a previous study reported, at least 65 residences with animal hoarding cases within Curitiba city limits were found, corresponding to 724 dogs [33]. A random sample calculation with a confidence level of 95% and an accuracy of 5% resulted in a minimum sampling of 251 dogs, sampling the largest possible number of dogs in each household for convenience, to assess the brucellosis seroprevalence in this population.

Blood sampling of dogs was carried out in 2017 at each residence and officially conducted by the City Secretary of the Environment. For legal reasons, human blood sampling was conducted in 2019, officially conducted by the City Secretary of Health.

All individuals were first informed about this study, invited to voluntarily participate, signed an informed consent form, responded to an epidemiological questionnaire, and then were subjected to blood sampling through cephalic puncture by certified nurses. Blood was collected from the dogs after the owners provided consent through jugular puncture by certified veterinarians.

### Epidemiological data and statistical analysis

For statistical purposes and due to exposure to the same accumulation conditions, all people and dogs residing in the households were considered hoarders and hoarding dogs, respectively, regardless of the exposure period prior to sampling. Signed informed consent was obtained before sampling by signing the Owner's Consent Term, with all procedures

performed in compliance with the National Human Ethics Research Committee for the use of human data and samples.

## Serological analyses

**2-Mercaptoethanol microplate agglutination test (2ME-MAT).** Dog and human serum samples were screened for anti-*B. canis* antibodies by the 2-mercaptoethanol microplate agglutination test (2ME-MAT) according to the French Agency for Food, Environmental and Occupational Health & Safety (ANSES, Maisons-Alfort, France) protocol, which is the national/European Union/WOAH (World Organization for Animal Health) reference laboratory for brucellosis [34]. The *B. canis* antigen 2ME-MAT was an inactivated suspension of *B. canis* that contained 4.5% cellules and 0.5% formaldehyde and was derived from the *B. canis* M-strain (nonmucoid strain). Serial dilutions from 1:20 to 1:640 were applied, with serum samples considered positive if antibody titers were ≥ 20 for *B. canis* [17], and final titers were determined at the last dilution with complete agglutination.

**Lateral Flow Immunoassay (LFIA).** Serum samples seropositive for 2ME-MAT were tested by a commercial lateral flow immunoassay (LFIA) for the qualitative detection of *B. canis* antibodies (Antigen Rapid C. *Brucella* Ab Test Kit, BioNote, Inc., Republic of Korea) [35,36].

## Molecular analyses

The seropositive samples for anti-*B. canis* antibodies were also molecularly tested by conventional PCR. DNA from each blood sample was extracted using a commercially available kit (Relia Prep gDNA Tissue Miniprep System–Promega, USA) and tested by conventional PCR for *Brucella* spp. detection using a standard protocol (ITS66: ACA TAG ATC GCA GGC CAG TCA and ITS279: AGA TAC CGA CGC AAA CGC TAC), as previously described [8]. After PCR, the samples were analyzed in 1.5% agarose gel by electrophoresis and stained with SYBR Safe DNA gel stain. DNA bands were visualized under UV light.

## Results

Although all 65 residences were visited within a year, dog sampling was allowed by residents only for 21 households, for a total of 264 dog samples. All people living in the 21 households were approached; however, blood samples were available from only 19 people from 11 households, and these individuals were fully evaluated.

No individual with AHD (0/19) was seropositive according to 2ME-MAT for anti-*B. canis* antibodies. Seropositivity for anti-*B. canis* antibodies by 2ME-MAT were observed in 14/264 (5.3%; 95% CI: 2.6–8.0%) dogs, with titers ranging from 20 to 640. At least one seropositive dog was found in 5/21 (23.8%, 95% CI: 5.6–42.0%) households, with the proportion of positive serological dogs per household ranging from 2/17 (11.8%) to 6/16 (37.5%).

All 14 dogs seropositive according to the 2ME-MAT were tested by LFIA, with 12/14 (85.7%) remaining seropositive. Thus, considering the two techniques (2ME-MAT and LFIA) together as a positive serological diagnosis, 12/264 (4.5%, 95% CI = 2.0–7.0%) dogs were seropositive in 4/21 (19.0%, 95% CI = 0.2–36.0%) households. These 14 seropositive samples were also tested by conventional PCR for *Brucella* spp., with no positive reaction (Table 1).

The majority of the tested individuals with AHD were women (13/19), most of whom were elderly (9/13), with ages ranging from 66 to 86 years.

Overall, seropositive dog samples were equally distributed between males and females (six each), with ages ranging from 1 to 16 years and dogs mostly living freely in the yard (9/12). In one household, two seropositive dogs lived in a collective kennel (Table 2).

**Table 1. Detection of anti-*B. canis* antibodies by 2ME-MAT and LFIA and *Brucella* spp.** DNA by PCR in human and dog samples.

| Diagnostic Test | | Samples | |
|---|---|---|---|
| | | Humans (n = 19) | Dogs (n = 264) |
| 2ME-MAT | Negative (%) | 19 (100) | 250 (94.7) |
| | Positive (%) | 0 (0) | 14 (5.3) |
| LFIA | Negative (%) | NE | 252 (95.5) |
| | Positive (%) | NE | 12 (4.5) |
| PCR | Negative (%) | NE | 264 (100) |
| | Positive (%) | NE | 0 (0) |

NE: not evaluated

## Discussion

The present study is the first comprehensive investigation of *B. canis* exposure in individuals with animal hoarding disorder (AHD) and their dogs. Considered a challenging diagnosis, *B. canis* exposure should be ideally detected by an association of different direct and indirect tests [13,17,26,28,37], such as 2ME-MAT and LFIA performed in the present study. Despite the reports of bacterial isolation as the gold positive standard and its use for early detection (2 weeks after bacterial contact), diagnosis requires laboratories with biosafety level 3, high cost, and high handling risk [15,16,38]. Although no diagnostic test has been considered solely satisfactory for assessing *B. canis* exposure, 2ME-MAT may be the choice for screening large populations [13,21], as it provides positive results 2–4 weeks after exposure [39]. For humans, we used the same diagnostic tests, as there are no approved tests for the diagnosis of *B. canis* to date [26].

Serological diagnosis of *B. canis* exposure may be even more relevant in this study, as individuals with AHD may experience self-health degradation [40] and poor perception of their animal and environmental health deterioration. Despite living in close contact with animals, clinical signs of dog brucellosis, such as abortion, may be ignored, unnoticed and unreported by elderly individuals with AHD, as reported in other low-income populations [41,42]. Although all individuals with AHD tested in our study were seronegative for *Brucella* spp., human exposure and outbreaks caused by *B. canis* have been reported [24,43]. A previous study revealed 4.6% (8/174) of individuals seropositive for *B. canis* from the slum community of Salvador, northeastern Brazil, related to the ongoing increase in the stray dog population [43]. A *B. canis* outbreak involving a female dog with her three puppies and three children and three adults was reported in Argentina in 2009. Humans exhibited fever, hepatomegaly and splenomegaly after close daily contact with the animals [24]. In addition, an unapparent *B. canis* infection was reported in kennel employees, suggesting the occupational risk of exposure [44]. Despite the positive outcome, human serological results should be cautiously interpreted, as the diagnostic techniques may vary and not be standardized, even in dogs [26].

The prevalence of anti-*B. canis* antibodies in dogs found in this study (4.5%) was within the range of those in previous surveys, from 5/175 (2.9%) [27] to 7/106 (6.6%) [28], which are all much lower than the 23/32 (71.8%) found in São Paulo state [30]. The last study focused on a population with a history of miscarriages, a history of failed conceptions, and characteristics of canine brucellosis [30]. As expected, the 4.5% dog overall seropositivity was lower than the per household seropositivity (11.8% to 37.5%), which may indicate that large unassisted dog populations living in confined places contributed to the transmission of *B. canis*, similarly observed in overpopulated kennels [17,30]. Furthermore, despite the absence of seropositivity in individuals with AHD and the comparatively low seroprevalence in dogs in the present study, *B.*

**Table 2. Distribution and frequency of presence of anti-*B. canis* antibodies in dogs and epidemiological data from dogs and individuals with AHD per household case.**

| Household case (n = 21) | Tested dogs (2017)* | Dogs per household (2019)** | Age (years) mean ± SD | Sex* | | Seropositive dogs (% per household) | Seronegative dogs (% per household) | Living place of seropositive dogs | Tested humans (2019)* | Age (years) | Gender |
|---|---|---|---|---|---|---|---|---|---|---|---|
| 1 | 13 | 20 | 4 ± 3 | Male | 2 | 0 (0.0) | 2 (15.4) | - | 2 | 66; 83 | Female; Female |
|  |  |  |  | Female | 11 | 0 (0.0) | 11 (84.6) |  |  |  |  |
| 2 | 16 | 29 | 3 ± 2 | Male | 8 | 0 (0.0) | 8 (50.0) | - | 3 | 16; 34; 69 | Female; Male; Male |
|  |  |  |  | Female | 8 | 0 (0.0) | 8 (50.0) |  |  |  |  |
| 3 | 3 | NA | 4 ± 2 | Male | 2 | 0 (0.0) | 2 (66.7) | - | NA | NA | NA |
|  |  |  |  | Female | 1 | 0 (0.0) | 1 (33.3) |  |  |  |  |
| 4 | 11 | 14 | 10 ± 0 | Male | 5 | 0 (0.0) | 5 (45.5) | - | 2 | 68; 86 | Female; Female |
|  |  |  |  | Female | 6 | 0 (0.0) | 6 (54.5) |  |  |  |  |
| 5 | 16 | NA | 3 ± 2 | Male | 7 | 3 (18.8) | 4 (25) | unleashed, leashed | NA | NA | NA |
|  |  |  |  | Female | 9 | 3 (18.8) | 6 (37.5) |  |  |  |  |
| 6 | 10 | NA | 4 ± 4 | Male | 6 | 2 (20.0) | 4 (40.0) | unleashed | NA | NA | NA |
|  |  |  |  | Female | 4 | 0 (0.0) | 4 (40.0) |  |  |  |  |
| 7 | 16 | 28 | 5 ± 3 | Male | 6 | 0 (0.0) | 6 (37.5) | - | 3 | 81; NA; NA | Female; Female; Male |
|  |  |  |  | Female | 10 | 0 (0.0) | 10 (62.5) |  |  |  |  |
| 8 | 18 | 30 | 4 ± 3 | Male | 10 | 0 (0.0) | 10 (55.6) | - | 1 | 59 | Female |
|  |  |  |  | Female | 8 | 0 (0.0) | 8 (44.4) |  |  |  |  |
| 9 | 12 | NA | 3 ± 3 | Male | 7 | 0 (0.0) | 7 (58.3) | - | NA | NA | NA |
|  |  |  |  | Female | 5 | 0 (0.0) | 5 (41.7) |  |  |  |  |
| 10 | 16 | 20 | 6 ± 3 | Male | 5 | 0 (0.0) | 5 (31.2) | - | 1 | 67 | Female |
|  |  |  |  | Female | 11 | 0 (0.0) | 11 (68.8) |  |  |  |  |
| 11 | 10 | NA | 5 ± 3 | Male | 1 | 0 (0.0) | 1 (10.0) | - | NA | NA | NA |
|  |  |  |  | Female | 9 | 0 (0.0) | 9 (90.0) |  |  |  |  |
| 12 | 11 | 13 | 5 ± 2 | Male | 6 | 0 (0.0) | 6 (54.5) | - | 1 | 71 | Female |
|  |  |  |  | Female | 5 | 0 (0.0) | 5 (45.5) |  |  |  |  |
| 13 | 9 | NA | 8 ± 6 | Male | 0 | 0 (0.0) | 0 (0.0) | - | NA | NA | NA |
|  |  |  |  | Female | 9 | 0 (0.0) | 9 (100.0) |  |  |  |  |
| 14 | 16 | NA | 5 ± 1 | Male | 12 | 0 (0.0) | 12 (75.0) | - | NA | NA | NA |
|  |  |  |  | Female | 4 | 0 (0.0) | 4 (25.0) |  |  |  |  |
| 15 | 14 | 20 | 5 ± 3 | Male | 5 | 0 (0.0) | 5 (35.7) | - | 1 | 60 | Male |
|  |  |  |  | Female | 9 | 0 (0.0) | 9 (64.3) |  |  |  |  |
| 16 | 9 | NA | 7 ± 5 | Male | 3 | 0 (0.0) | 3 (33.3) | - | NA | NA | NA |
|  |  |  |  | Female | 6 | 0 (0.0) | 6 (66.7) |  |  |  |  |
| 17 | 6 | NA | 7 ± 3 | Male | 3 | 0 (0.0) | 3 (50.0) | - | NA | NA | NA |
|  |  |  |  | Female | 3 | 0 (0.0) | 3 (50.0) |  |  |  |  |
| 18 | 17 | 30 | 5 ± 3 | Male | 5 | 0 (0.0) | 5 (29.4) | kenneled | 2 | 42; 67 | Female; Female |
|  |  |  |  | Female | 12 | 2 (11.8) | 10 (58.8) |  |  |  |  |
| 19 | 12 | NA | 4 ± 4 | Male | 3 | 0 (0.0) | 3 (25.0) | - | NA | NA | NA |
|  |  |  |  | Female | 9 | 0 (0.0) | 9 (75.0) |  |  |  |  |
| 20 | 10 | 15 | 13 ± 3 | Male | 6 | 1 (10.0) | 5 (50.0) | unleashed | 2 | 74; NA | Female; Male |
|  |  |  |  | Female | 4 | 1 (10.0) | 3 (30.0) |  |  |  |  |
| 21 | 19 | 30 | 3 ± 1 | Male | 7 | 0 (0.0) | 7 (36.8) | - | 1 | 73 | Male |
|  |  |  |  | Female | 12 | 0 (0.0) | 12 (63.2) |  |  |  |  |

NA: not available data. Households in bold allowed human blood sampling only in 2019.

* Dogs blood sampling was carried out in 2017 and human blood sampling was conducted in 2019, due to permission of city secretary of health and ethics committee.

** Total of dogs in each household was investigated only in 2019.

*canis* outbreaks should be considered in such human populations due to the high burden and recurrent character of *B. canis* circulation in high-density dog populations and the constant introduction of susceptible animals.

These findings in dogs highlight human vulnerability as a risk factor for brucellosis, as has been observed for other zoonotic diseases in homeless, indigenous, and quilombola communities [45–48]. In addition, negative results for individuals with AHD may indicate that *B. canis* has the lowest zoonotic potential in the genus *Brucella* [12,17]. Nonetheless, seropositive dogs in close and continuous contact with owners with AHD, mostly elderly, unassisted (last medical visit over 6 months prior), lonely and with comorbidities [1,6], may be a tragic combination.

Polymerase chain reaction (PCR) has been commonly used to confirm *Brucella* species [10] and as a useful confirmatory test in seropositive dogs [8,16,49]. However, nondetection of DNA in seropositive samples for anti-*B. canis* antibodies may be the result of intermittent bacteremia and intracellular characteristics, leading to low diagnostic sensitivity [15].

Hoarding disorder cases are challenging mostly due to neglect, the multiplicity and complexity of associated factors [5], and difficulty accessing affected populations [7]. A recent study in the Metropolitan Region of Curitiba, Paraná, indicated challenges faced by the City Hall official services regarding animal hoarding disorders, such as the lack of standardization and exchange of service information between the involved city secretaries, including health, environment, and social services [50]. As a consequence, cases become chronic, with worsening of accumulation and consequences, particularly zoonoses.

*B. canis* is transmitted mainly through contact with tissues and secretions from contaminated reproductive organs of dogs [11,13,16,19]. Although the pathogen has also been found in whole blood [16], this may not be the best sample for DNA detection by PCR. Thus, negative PCR results for blood samples in this study may not be indicative of the absence of infection.

As a limitation of this study, no sampling was performed for tissues, saliva, nasal or genital secretions, or urine, particularly following reproductive problems, which would have increased the diagnostic sensitivity [16]. Furthermore, access to and monitoring of households of individuals with AHD was limited by the classic "refusal to receive help" characteristic of the AHD population, which also impacted the low sampling of individuals with AHD in this study. Furthermore, the sampling dates used were different for dogs and individuals with AHD in compliance with different time permits from the ethics committee of animal use and the city secretary of health, which may have impacted the results due to potential dynamics over time in *B. canis* exposure. In addition, the epidemiological questionnaires failed to include important questions on associated risk factors for *B. canis* exposure, such as serology following reproductive failure or molecular testing of abortion tissues.

## Conclusion

The present study is the first to assess brucellosis in a vulnerable population as a silent disease based on the environment and susceptible population. Despite the absence of seropositivity in individuals with AHD and the comparatively low seroprevalence in dogs, *B. canis* circulation and outbreaks should be considered in such human populations due to the high burden and recurrent character of *B. canis* exposure in high-density dog populations and the constant introduction of susceptible animals. In conclusion, serological and molecular assessments of vulnerable populations, particularly those with silent diseases, may serve as the basis for public health policies, including systematic diagnosis, control, monitoring, and prevention.

## Supporting information

**S1 Questionnaire. Epidemiological questionnaire used to investigate serological and molecular *Brucella canis* and associated factors in individuals with animal hoarding disorder and their dogs in Curitiba.**
(DOCX)

## Acknowledgments

The authors are deeply thankful to professionals from the Curitiba Secretaries of Health, the Environment and Social Assistance.

## Author Contributions

**Conceptualization:** Letícia Schiavo, Meila Bastos de Almeida, Jane Megid, Alexander Welker Biondo.

**Data curation:** Letícia Schiavo, Matheus Lopes Ribeiro, Graziela Ribeiro da Cunha, Carlos Eduardo de Santi, Louise Bach Kmetiuk, Jane Megid, Alexander Welker Biondo.

**Formal analysis:** Letícia Schiavo, Matheus Lopes Ribeiro, Jane Megid.

**Funding acquisition:** Alexander Welker Biondo.

**Investigation:** Alexander Welker Biondo.

**Methodology:** Letícia Schiavo, Claire Ponsart, Jane Megid, Alexander Welker Biondo.

**Project administration:** Alexander Welker Biondo.

**Resources:** Letícia Schiavo, Jane Megid, Alexander Welker Biondo.

**Software:** Letícia Schiavo.

**Supervision:** Meila Bastos de Almeida, Alexander Welker Biondo.

**Validation:** Acácia Ferreira Vicente, Claire Ponsart, Jane Megid.

**Visualization:** Alexander Welker Biondo.

**Writing – original draft:** Letícia Schiavo, Matheus Lopes Ribeiro, Meila Bastos de Almeida, Graziela Ribeiro da Cunha, Giselle Almeida Nocera Espírito Santo, Vivien Midori Morikawa, Acácia Ferreira Vicente, Claire Ponsart, Carlos Eduardo de Santi, Louise Bach Kmetiuk, Jane Megid, Alexander Welker Biondo.

**Writing – review & editing:** Letícia Schiavo, Meila Bastos de Almeida, Louise Bach Kmetiuk, Jane Megid, Alexander Welker Biondo.

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
