## [Decision Letter · Decision Letter 0]

22 Nov 2023

Dear Professor Megid,

Thank you very much for submitting your manuscript "One Health approach on Brucella canis: serological and molecular detection in animal hoarding individuals and their dogs" for consideration at PLOS Neglected Tropical Diseases. As with all papers reviewed by the journal, your manuscript was reviewed by members of the editorial board and by several independent reviewers. In light of the reviews (below this email), we would like to invite the resubmission of a significantly-revised version that takes into account the reviewers' comments. 

We cannot make any decision about publication until we have seen the revised manuscript and your response to the reviewers' comments. Your revised manuscript is also likely to be sent to reviewers for further evaluation.

Sincerely,

Joseph M. Vinetz

Section Editor

Joseph Vinetz

Section Editor

Reviewer's Responses to Questions

**Key Review Criteria Required for Acceptance?**

**Methods**

-Are the objectives of the study clearly articulated with a clear testable hypothesis stated?

-Is the study design appropriate to address the stated objectives?

-Is the population clearly described and appropriate for the hypothesis being tested?

-Is the sample size sufficient to ensure adequate power to address the hypothesis being tested?

-Were correct statistical analysis used to support conclusions?

-Are there concerns about ethical or regulatory requirements being met?

Reviewer #1: -Are the objectives of the study clearly articulated with a clear testable hypothesis stated? yes

-Is the study design appropriate to address the stated objectives? yes

-Is the population clearly described and appropriate for the hypothesis being tested? no

-Is the sample size sufficient to ensure adequate power to address the hypothesis being tested? no

-Were correct statistical analysis used to support conclusions? no

-Are there concerns about ethical or regulatory requirements being met? yes

**Results**

-Does the analysis presented match the analysis plan?

-Are the results clearly and completely presented?

-Are the figures (Tables, Images) of sufficient quality for clarity?

Reviewer #1: -Does the analysis presented match the analysis plan? yes

-Are the results clearly and completely presented? no

-Are the figures (Tables, Images) of sufficient quality for clarity? no

**Conclusions**

-Are the conclusions supported by the data presented?

-Are the limitations of analysis clearly described?

-Do the authors discuss how these data can be helpful to advance our understanding of the topic under study?

-Is public health relevance addressed?

Reviewer #1: -Are the conclusions supported by the data presented? partly

-Are the limitations of analysis clearly described? yes

-Do the authors discuss how these data can be helpful to advance our understanding of the topic under study? no

-Is public health relevance addressed? yes

**Editorial and Data Presentation Modifications?**

Reviewer #1: (No Response)

**Summary and General Comments**

Reviewer #1: The study has serious limitations.

Firstly, the human samples are insufficient, while the dog samples are incompletely described. The authors should include a comprehensive table that includes number of dogs present in the household, number of dogs sampled, number of people living in the household and number of sampled people (including sex and age)

Secondly, the authors mention questionnaires both in methods and discussion, but do not provide any data in the results section. Was there any statistical analysis performed on them? The questionnaire should be included as supplementary material

Seropositivity does not in any way indicate an active disease.

Finally, a thorough revision of English language is required

PLOS authors have the option to publish the peer review history of their article (what does this mean?). If published, this will include your full peer review and any attached files.

Reviewer #1: No
---

## [Decision Letter · Decision Letter 1]

6 Feb 2024

Dear Dr Biondo,

We are pleased to inform you that your manuscript 'One Health approach for Brucella canis: Serological and molecular detection in animal-hoarding individuals and their dogs' has been provisionally accepted for publication in PLOS Neglected Tropical Diseases.

Best regards,

Stuart D. Blacksell

Section Editor

Joseph Vinetz

Section Editor

---

## [Editor Report · Acceptance letter]

7 Mar 2024

Dear Dr Biondo,

We are delighted to inform you that your manuscript, "One Health approach for <i>Brucella canis<i>: Serological and molecular detection in animal-hoarding individuals and their dogs," has been formally accepted for publication in PLOS Neglected Tropical Diseases.

Best regards,

Shaden Kamhawi

co-Editor-in-Chief

Paul Brindley

co-Editor-in-Chief
